# Role of Long Noncoding RNAs ZlMSTRG.11348 and UeMSTRG.02678 in Temperature-Dependent Culm Swelling in *Zizania latifolia*

**DOI:** 10.3390/ijms22116020

**Published:** 2021-06-02

**Authors:** Zheng-Hong Wang, Ning Yan, Xi Luo, Sai-Sai Guo, Shu-Qin Xue, Jiang-Qiong Liu, Shen-Shen Zhang, Li-Wen Zheng, Jing-Ze Zhang, De-Ping Guo

**Affiliations:** 1Department of Horticulture, College of Agriculture and Biotechnology, Zhejiang University, Hangzhou 310058, China; e13858121328@163.com (Z.-H.W.); 21816049@zju.edu.cn (X.L.); melodygss@163.com (S.-S.G.); xsq@zju.edu.cn (S.-Q.X.); jqliu169@zju.edu.cn (J.-Q.L.); 21916127@zju.edu.cn (S.-S.Z.); 13379806160@163.com (L.-W.Z.); 2Tobacco Research Institute of Chinese Academy of Agricultural Sciences, Qingdao 266101, China; yanning@caas.cn; 3Institute of Biotechnology, College of Agriculture and Biotechnology, Zhejiang University, Hangzhou 310058, China

**Keywords:** long noncoding RNA, temperature, gall formation, amino acid metabolism, plant defence response, *Ustilago esculenta*, *Zizania latifolia*

## Abstract

Temperature influences the physiological processes and ecology of both hosts and endophytes; however, it remains unclear how long noncoding RNAs (lncRNAs) modulate the consequences of temperature-dependent changes in host–pathogen interactions. To explore the role of lncRNAs in culm gall formation induced by the smut fungus *Ustilago esculenta* in *Zizania latifolia*, we employed RNA sequencing to identify lncRNAs and their potential *cis*-targets in *Z. latifolia* and *U. esculenta* under different temperatures. In *Z. latifolia* and *U. esculenta*, we identified 3194 and 173 lncRNAs as well as 126 and four potential target genes for differentially expressed lncRNAs, respectively. Further function and expression analysis revealed that lncRNA ZlMSTRG.11348 regulates amino acid metabolism in *Z. latifolia* and lncRNA UeMSTRG.02678 regulates amino acid transport in *U. esculenta*. The plant defence response was also found to be regulated by lncRNAs and suppressed in *Z. latifolia* infected with *U. esculenta* grown at 25 °C, which may result from the expression of effector genes in *U. esculenta*. Moreover, in *Z. latifolia* infected with *U. esculenta*, the expression of genes related to phytohormones was altered under different temperatures. Our results demonstrate that lncRNAs are important components of the regulatory networks in plant-microbe-environment interactions, and may play a part in regulating culm swelling in *Z. latifolia* plants.

## 1. Introduction

The smut fungus *Ustilago esculenta* infects the host *Zizania latifolia* plant causing formation of culm gall [1,2,3]. Apart from culm hypertrophy, which represents the characteristic symptom associated with infection of this fungus, no apparent change appears in *Z. latifolia* plants [2]. In maize plants, *Ustilago maydis* generally induces tumours once it invades plant tissues [4], however tumours initiation by *U. esculenta* is temperature dependent, and tends to occur in seasons with temperature ranging from 21 °C to 27 °C under field conditions. When plants grow at temperatures above 30 °C or below 15 °C, swollen galls do not develop though *U. esculenta* infection [5,6]. To date, the underlying mechanism responsible for this obvious difference in gall formation dynamics between thermal conditions is unknown.

As a fungus endophyte, *U. esculenta* completes its life cycle in plant tissues, leading to formation of culm gall [3,7]. In terms of the ‘disease triangle,’ disease outbreaks occur only under proper environmental conditions following pathogen colonisation and proliferation in host plants. Among the environmental factors, temperature, which generally influences the physiology, ecology, and growth of hosts and microorganisms, is considered to be integral to the promotion of disease outbreaks in plants [8].

Moreover, temperature can significantly affect the interactions between plants and microbes [9]. For instance, in *Arabidopsis thaliana*, cold leads to derepression of salicylic acid (SA) immunity under pathogen attack, thereby accumulating SA and pathogenesis-related proteins, a response that becomes repressed at optimal temperatures [10,11]. Further, altered plant performance in response to temperature ranges in biotic environments can alter the disease severity and even microbe phenotype [8]. Interestingly, the behaviours of microorganisms may differ across temperatures in plant-microbe interaction due to dimorphic transition of fungi for pathogenesis [12,13,14], inducing non-pathogenic attacks and decreased virulence protein stability [15,16]. Although pronounced changes regarding plant-microbe interactions in response to temperature have been described, the underlying mechanisms remain elusive.

Amino acids, constituents of proteins, function in many biological processes [17]. Some amino acids have been shown to play important roles in tumour progression. For example, serine and glycine contribute to tumour growth [18], asparagine and serine to breast cancer [19,20], glutamine to the growth and metabolism of tumour cells [21]; arginine, lysine, leucine, methionine and isoleucine are reportedly required for tumour formation induced by *U. maydis* in maize plants [22].

Amino acids also fuel the function of microorganisms during interactions with their host. For instance, methionine is required for *Magnaporthe oryzae* in appressoria formation to facilitate penetration of host leaves [23]. Additionally, amino acids take part in nitrogen-fixing symbiosis [24,25,26,27], while, threonine, isoleucine, and methionine are involved in response to heat shock stress in fungi [28]. However, the mechanisms driving these responses remain largely unclear. The precise role played by amino acids in plant-microbe-environment interactions requires further investigation.

Long noncoding RNAs (lncRNAs) are RNAs longer than 200 bp without apparent coding potential. They are classified as long intergenic noncoding RNAs, intronic noncoding RNAs and natural antisense transcripts. LncRNAs regulate gene expression through sequence complementarity or homology with RNAs, forming molecular frames for macromolecular complexes [29]. In plants, lncRNAs function in plant responses to abiotic and biotic stresses and are reportedly involved in modulating salt stress [30], immunity in *A. thaliana* against *Fusarium oxysporum* [31] and *Pseudomonas syringe* [32], plant cold acclimation [33], as well as plant response to high temperature stress [34]. A striking regulatory role for lncRNAs in tumour formation has also been extensively documented in animals and humans, including roles in oral squamous cell carcinoma tumorigenesis [35] and gastric cancer [36]. Indeed, lncRNAs have been evaluated in the progression of several cancers in animals; however, no study has yet reported on whether lncRNAs are directly involved in plant tumour formation.

The formation of *U. esculenta*-induced culm gall in *Z. latifolia* is accompanied by specific biochemical and physiological changes, such as decreased activities of catalase and superoxide dismutase [37], enhanced photosynthesis [2], altered sugar metabolism [38], and induced cytokinin and indole acetic acid production [3]. Similarly, in tumours induced by *U. maydis* in maize, cytokinin production and nitrogen transport are promoted [4,39]. Although *U. maydis*-maize interactions have been investigated in the context of tumour formation [4], the molecular mechanisms associated with temperature effects on plant tumour formation still remain unknown.

Herein, we aimed to analyse the whole transcriptome of *Z. latifolia* infected with *U. esculenta* at different temperatures, resulting in the identification of lncRNAs and genes relevant to culm gall formation. These findings might provide new insights into how lncRNAs contribute to plant tumour formation, and an inventory of lncRNA for future studies in plant-microbe-environment interactions.

## 2. Results

### 2.1. Plant Culm Morphology

The appearance and morphology of plant culms are shown in Figure 1. *Z. latifolia* plants infected by *U. esculenta* (JB) produced a swollen culm gall at 25 °C (JB25). 

The other plants, including *Z. latifolia* plants without *U. esculenta* infection (CK) grown at 25 °C (CK25), as well as CK and JB plants grown at 35 °C (CK35, JB35), did not produce swollen culm gall, and exhibited a significant decrease in length, weight, perimeter, and volume in culms compared to JB25 (*p* < 0.05). These results clearly demonstrate that culm gall can form in JB plants grown at 25 °C, while gall induction was completely suppressed by a temperature of 35 °C.

### 2.2. Identification and Characterisation of lncRNAs in Z. latifolia and U. esculenta

To develop a comprehensive catalogue of lncRNAs, whole transcriptome strand-specific RNA sequencing was performed for JB and CK plants grown at 25 °C and 35 °C. Overall, 3194 and 173 lncRNAs were identified in *Z. latifolia* and *U. esculenta*, respectively. The length of the transcript and exon counts as well as the expression level of the putative lncRNAs were compared with those of mRNA identified in *Z. latifolia* and *U. esculenta* (Appendix A). The length distribution of the identified lncRNAs was shorter than that of the mRNAs, while the exon number of lncRNAs was fewer than that of the mRNAs. The expression level of each transcript was then estimated using fragments per kilobase of exon per million mapped reads (FPKM), and the overall expression level of lncRNAs was lower than that of the mRNA.

### 2.3. lncRNA and mRNA Expression in Z. latifolia and U. esculenta

RNA-sequencing was performed on culms of JB and CK plants grown at 25 °C and 35 °C. Our analysis indicated that 3194 and 173 lncRNAs were identified in *Z. latifolia* and *U. esculenta*, respectively. Among those lncRNAs identified in *Z. latifolia*, 144 were uniquely expressed in JB25 (Figure 2a). Meanwhile, among the lncRNAs identified in *U. esculenta*, 106 were uniquely expressed in JB25 (Figure 2b). Further analysis showed that 293 differentially expressed (DE) lncRNAs were identified in *Z. latifolia* between JB35 and JB25 (90 upregulated in JB25) (Figure 2c), and 149 between CK35 and CK25 (134 upregulated in CK25), respectively (Figure 2d). In *U. esculenta*, only ten DE lncRNAs were identified between JB25 and JB35, five of which were upregulated in JB25 (Figure 2e).

In total, 21,832 plant genes and 5567 fungal genes were identified in CK and JB plants, respectively (Figure 2f,g). In *Z. latifolia*, 7036 and 6549 DE genes were identified between JB25 and JB35 (Figure 2h) and between CK25 and CK35 (Figure 2i), respectively. Specifically, in *U. esculenta*, 301 DE genes were identified between JB25 and JB35, including genes encoding effectors and transporters, genes involved in protein synthesis and amino acid metabolism (Figure 2j, Appendix A).

### 2.4. Functional Analysis of lncRNAs

To analyse the functions of lncRNAs, potential *cis*-targets of lncRNAs were predicted. In *Z. latifolia*, 126 potential *cis*-target genes of DE lncRNAs were identified (Appendix A), and 91 potential *cis*-target genes were identified for lncRNAs uniquely expressed in JB25 (Appendix A), whereas only four potential *cis*-target genes of DE lncRNAs were identified in *U. esculenta* (Table 1, Figure 3a). 

Further, the 126 DE lncRNAs in *Z. latifolia* and their potential *cis*-target genes were divided into three groups (G1–G3) according to their expression patterns under different temperatures (Figure 3b). In group G1, there were 76 lncRNAs differentially expressed between JB25 and JB35, indicating that these lncRNAs, and their potential *cis*-target genes, may be involved in temperature-dependent gall formation. Meanwhile, in group G2, 31 DE lncRNAs were found between CK25 and CK35, implying that they, along with their potential *cis*-target genes, are not associated with temperature-dependent gall formation. In addition, in group G3, 19 DE lncRNAs were identified between both JB25/JB35 and CK25/CK35, suggesting that some of these lncRNAs and their potential *cis*-target genes may be involved in temperature-dependent gall formation.

### 2.5. Plant Defence Response

Among the DE lncRNAs in G1 and G3 groups, lncRNA ZlMSTRG.30807 was identified, which may regulate the expression of the disease resistance protein RPM1 (*Zl*.00376) (Appendix A). Interestingly, lncRNA ZlMSTRG.30807 was downregulated in JB25 compared to JB35, and the expression of its *cis*-target gene *ZlRPM1* was reduced by more than 100-fold compared to JB35, suggesting that lncRNA ZlMSTRG.30807 was positively correlated with *ZlRPM1* regulation, and the interaction between the lncRNA ZlMSTRG.30807 and *ZlRPM1* gene was further predicted (Figure 4a). Moreover, we identified 229 DE genes related to plant defence response between JB25 and JB35 and between CK25 and CK35 (Figure 4b, Appendix A). Most of these DE genes (138 of 229) were downregulated in JB25 compared to JB35, including genes encoding calcium-dependent protein kinase (*Zl*.22932, *Zl*.16176, *Zl*.11318), WRKY transcription factor 29 (*Zl*.00792, *Zl*.20037, *Zl*.16091), senescence-induced receptor-like serine/threonine-protein kinase FRK1 (*Zl*.02687), disease resistance protein RPM (*Zl*.00376, *Zl.*06039), heat shock protein (*Zl*.22886), and disease resistance protein RPS (*Zl*.06296). Only 74 DE genes were identified between CK25 and CK35, most of which (64 of 74) were upregulated in CK25 (Appendix A).

To validate the putative relationship between lncRNA ZlMSTRG.30807 and *ZlRPM1*, their expression levels were examined by quantitative reverse-transcription PCR (qRT-PCR) (Figure 4c). qRT-PCR results were consistent with the RNA-seq data for expression of lncRNA and mRNA (Appendix A). Additionally, expression levels of the pattern-triggered immunity (PTI) marker genes *ZlWRK29* and *ZlFRK1*, as well as *UePEP1*, which encodes the effector pep1 in *U. esculenta*, were also examined by qRT-PCR. Through qRT-PCR analysis, we observed downregulation of *ZlWRK29* and *ZlFRK1* in JB25 compared to JB35, and slight difference between CK25 and CK35. Moreover, increased expression of *UePEP1* was observed in JB25 compared to JB35 (Figure 4d,e).

Additionally, the expression pattern of *ZlRPM1* in *Z. latifolia* differed from that in *A. thaliana* [40]. Thus, sequence alignment and phylogenetic analysis were further performed (Figure 5). Phylogenetic analysis showed that *ZlRPM1* is homologous with RPM from other plant species, including *Oryza brachyantha*, *Oryza sativa*, and *Sorghum bicolor*. Further, results of amino acid sequence alignment revealed that a portion of the ZlRPM1 amino acid sequence was lost compared with that of RPM1 from other plant species.

### 2.6. Plant Hormones

Among the DE lncRNAs in G1 and G3 groups, we found that the *cis*-target genes of ZlMSTRG.09543, ZlMSTRG.05484, and ZlMSTRG.09415 were related to ethylene (ET) signalling pathway (Appendix A). Using the RNAplex tool, we found that lncRNA ZlMSTRG.09543 could interact with the ethylene-responsive transcription factor 1 (*ZlETF1)* gene (*Zl*.05881) (Figure 6a). The ZlMSTRG.09543 was upregulated in JB25 compared to JB35, but no significant difference was observed between CK25 and CK35. Further, *ZlETF1* showed a slightly increased expression in JB25 compared to JB35, and a lower expression in CK25 compared to CK35. In addition, it was predicted that lncRNA ZlMSTRG.09415 might interact with AP2-like ethylene-responsive transcription factor AIL5 (*ZLAIL5*) (*Zl*.25034) (Figure 6b). ZlMSTRG.09415 was downregulated in JB25 compared to JB35. Similarly, *ZlAIL5* exhibited lower expression in JB25 than in JB35, while no significant difference between CK25 and CK35 (Appendix A). Among six DE genes involved in ET biosynthesis identified in *Z. latifolia* (Table 2), gene encoding ethylene-forming enzyme 1-aminocyclopropane-1-carboxylate oxidase (*ZlACO*) (*Zl*.07467, *Zl*.09320) was downregulated in JB25 compared to JB35, but only slightly different between CK25 and CK35.

To further investigate the role of ET in temperature-dependent gall formation, qRT-PCR was performed to measure the expression of ZlMSTRG.09543, *ZlETF1*, and *ZlACO* in *Z. latifolia* (Figure 6c). The qRT-PCR results were consistent with our RNA-seq data (Appendix A). *ZlACO* gene showed a significantly lower expression in JB25 compared to JB35, while slightly lower expression in CK25 compared to CK35.

Furthermore, DE genes involved in cytokinin metabolism were identified in *Z. latifolia* (Table 2). Two of three genes (*Zl.*15021 and *Zl.*20687) encoding isopentenyltransferase, the key enzyme in cytokinin biosynthesis, were upregulated in JB25 compared to JB35. In addition, two of three genes (*Zl*.05044 and *Zl*.12768) encoding cytokinin dehydrogenase, an enzyme responsible for cytokinin degradation, were downregulated in JB25 compared to JB35; however, the expression of genes associated with cytokinin metabolism in CK25 differed only slightly from those in CK35.

### 2.7. Amino Acid Metabolism

To assess whether amino acids involve in the interaction between *U. esculenta* and its host plant *Z. latifolia*, lncRNAs with its *cis*-target genes related to amino acid metabolism were identified. Among these lncRNAs, ZlMSTRG.11348 was uniquely expressed in *Z. latifolia* in JB25, its *cis*-target gene may be bark storage protein A, *ZlBSP* (Appendix A, Figure 7a). We also found *ZlBSP* was downregulated in JB25 compared to JB35, while no significant difference between CK25 and CK35. Moreover, 194 DE genes involved in amino acid metabolism were identified in *Z. latifolia*, among which 18, 11, and four genes were involved in glutamate, glycine, and histidine metabolism, respectively (Appendix A).

In *U. esculenta*, DE lncRNAs UeMSTRG.02678 was also identified, and predicted to interact with the gene encoding major facilitator superfamily transporter, *UeMFS* (Table 1, Figure 7b). Both UeMSTRG.02678 and *UeMFS* were upregulated in JB25 compared to JB35.

The expression of genes of probable sarcosine oxidase, *ZlPSO*, histidinol dehydrogenase, *ZlHID*, glutamate synthase, *ZlGLS*, glutamine synthetase, *ZlGNS*, and aldehyde dehydrogenase, *ZlADG* in *Z. latifolia* as well as lncRNA UeMSTRG.02678, *UeMFS*, glycine decarboxylase, *UeGYD*, glutamate decarboxylase, *UeGUD*, and ribosomal 40S subunit, *UeR40* in *U. esculenta* were further examined by qRT-PCR (Figure 7c–e). The qRT-PCR results were matched with the RNA-seq data (Appendix A), and indicated that *ZlPSO*, *ZlHID*, *ZlGLS*, and *UeR40* were all highly induced in JB25 compared to JB35. Meanwhile, *ZlGNS*, *ZlADG*, *UeGYD*, and *UeGUD* genes were downregulated in JB25 compared to JB35. Further, a slightly altered expression of *ZlPSO*, *ZlHID*, *ZlGLS*, *ZlGNS*, and *ZlADG* genes was observed in CK25 in comparison to CK35.

To further investigate how amino acids influence the growth of *U. esculenta* under different temperatures, in vitro culture of *U. esculenta* with glutamate, glycine, and histidine were performed (Figure 7f). At 35 °C, the growth of *U. esculenta* in basic medium (BM) added with glycine and histidine was similar to that in control. Notably, the growth of *U. esculenta* in BM exhibited a significantly slower following addition of glutamate from 12 h to 72 h compared to control. However, at 25 °C, the growth of the fungus in BM with addition of amino acids was significantly faster from 48 h to 72 h compared to control.

## 3. Discussion

Temperature is a primary environmental factor that affects the growth and development of plants. At elevated temperatures (usually ≥ 30 °C), both developmental and growth parameters are affected [41], and growth of microbes in plants and outcome of plant–microbe interactions may be modified [7]. Temperature influences the formation of culm gall caused by *U. esculenta* in *Z. latifolia* [5]; however, the regulatory network that controls the expression of genes during culm gall development is not well studied. Whether lncRNAs act as a developmental regulator of culm gall remains unknown. In this study, we examined whether temperature play a role in activation or repression of genes associated with culm gall formation, and the role of lncRNAs in controlling differential expression of numerous genes during gall formation. Our results have revealed that a number of plant defence response, phytohormone, and amino acid metabolism-related genes are potentially controlled by temperatures in host and the fungus *U. esculenta*.

### 3.1. Culm Gall Formation Is Regulated by Temperature

Gall formation in *Z. latifolia* is significantly influenced by temperature, and it can initiate when temperatures fluctuate between 21 °C and 27 °C in the field [5]. In this study, we noticed a culm gall formation in *Z. latifolia* plants infected with *U. esculenta* at 25 °C; however, gall was not observed in control plants without *U. esculenta* infection, indicating that *U. esculenta* is the causal agent for culm gall formation (Figure 1). Interestingly, when *U. esculenta*-infected plants were grown at a high temperature (35 °C), no swollen gall was produced, demonstrating that gall formation in *U. esculenta*-infected plants appears to depend on temperature (Figure 1).

LncRNAs have been recognised as important regulators of gene expression in plant cell growth [42] and play a role in plant-microbe interactions [32]. Therefore, temperature may regulate the expression of lncRNAs and further modulate genes involved in plant physiological processes. For instance, heat alters the expression pattern of lncRNAs and regulate the expression of their downstream genes involved in specific biological pathways responding to stresses in *Brassica juncea* [34]. In this study, we, therefore, performed RNA sequencing of culm samples from *Z. latifolia* plants infected with or without *U. esculenta* infection grown at 35 °C or 25 °C. A total of 3194 and 173 lncRNAs were identified in *Z. latifolia* and *U. esculenta*, respectively (Figure 2).

### 3.2. Temperature Alters lncRNA-Mediated Genes Related to Plant Defence Response and Hormone

We could not find DE lncRNAs associated with cell division in *Z. latifolia;* however, we were able to detect DE lncRNAs that is involved in plant defence response (Figure 4). Specifically, in JB plants, lncRNA ZlMSTRG.30807 and its *cis*-target gene *ZlRMP1* were downregulated in JB25 at low temperature (25 °C) compared to JB35 at high temperature (35 °C). Contrarily, in CK plants, no significant difference in the expression of lncRNA ZlMSTRG.30807 and its *cis*-target gene *ZlRMP1*, was observed between CK25 and CK35. In addition, genes involved in plant defence responses were differentially expressed at different temperatures (Appendix A). Activation of plant defence response might possibly contribute to the low plant growth at higher temperature [43].

We observed DE genes involved in effector-triggered immunity (ETI) and PTI, including marker genes of PTI *ZLWRKY29* and *ZlFRK1*, indicating that ETI and PTI were largely compromised and that programmed cell death, hypersensitive response, ROS burst, and activation of the plant stress hormone regulatory network, were inhibited in JB under cooler temperature (Figure 4; Table 2) [44,45,46]. Consistently, the expression of *ZlAIL5*, regulated by DE lncRNA ZlMSTRG.09415, as well as genes involved in ET biosynthesis, was lower in JB25 than JB35 (Figure 6), which may cause suppression of ET biosynthesis [47] and downregulation of ET-regulated defence-related gene expression [48], further leading to inhibition of the ET-dependent disease resistance response in JB25 [49]. Moreover, upregulation of effector genes in *U. esculenta* may contribute to the inhibition of the defence response in JB under cooler temperatures (Figure 4, Appendix A) [50,51,52,53]. These results were consistent with a previous study [3], while different slightly from a report on *A. thaliana* [40]. The loss of a portion of the ZlRPM1 amino acid sequence caused by the long-term cultivation and artificial selection of this crop may account for this difference.

Our results showed that high temperature induces increased expression of *ZlRPM1* regulated by lncRNA ZlMSTRG.30807 and activation of plant defence response, possibly leading to suppressed fungal cell proliferation, whereas under cooler temperature, downregulation of ZlMSTRG.30807 results in an inhibition of defence response, which would not only provide the opportunity for *U. esculenta* to proliferate in plant tissues but also induces plant cell division [43]. Upregulation of genes associated with cytokinin biosynthesis and downregulation of cytokinin degradation genes were observed in JB25, suggesting that greater cytokinetic activity might involve in gall formation [3]. These results indicated that lncRNA differential expression driven by temperature variation might regulate the plant defence response in *Z. latifolia* infected with *U. esculenta*. However, the functional analysis of lncRNA ZlMSTRG.30807 in the regulation of plant defence responses requires further investigation.

### 3.3. Expression of Amino Acid Metabolism and Sugar Metabolism Genes Influences Gall Formation

Previous studies have shown that amino acids participate in many tumour pathways. For instance, restriction of serine and glycine drives the accumulation of toxic sphingolipids within tumours and reduces tumour cell growth [18]. Further, histidine deprivation reduce the growth of *Drosophila* tumour cells [54]. In order to examine whether lncRNAs are possibly involved in the amino acid mechanism at different temperatures, we identified genes involved in the amino acid metabolism from our RNA-seq data. We identified only a few DE genes involved in the amino acid biosynthesis between CK plants grown at low temperature (25 °C) and high temperature (35 °C), while the upregulation of *ZlPSO*, *ZlHID*, and *ZlGLS*, and downregulation of *ZlGNS* and *ZlADG* was observed in JB plants at low temperature (25 °C) compared to high temperature (35 °C) (Figure 7), suggesting that variation in ambient temperature scarcely affected the amino acid mechanism in CK plants, but promoted the synthesis of glycine, histidine and glutamate in *Z. latifolia* infected by *U. esculenta* at a cooler temperature [55,56,57].

Besides this, we also found lncRNA ZlMSTRG.11348 was uniquely expressed in JB plants at 25 °C and, thus, may result in a lower expression of *ZlBSP* in JB25 compared to JB35, indicating that amino acids synthesis was induced in *Z. latifolia* infected with *U. esculenta* under cooler temperatures [58]. Hence, these synthesized amino acids may serve as nitrogen sources for plant cell division during gall formation at cooler temperature.

Amino acids also represent the most favourable nitrogen sources assimilated by fungi for growth inside the plant environment [4]. For example, glutamate [25], glycine [24], and histidine [26] are indispensable for the establishment and maintenance of the interaction between microorganisms and their hosts. Also, high glycine and glutamic acid contents were detected in tumours induced by *U. maydis* [59]. We, therefore, determined whether these amino acids fuelled *U. esculenta* cell proliferation under cooler temperature. Our qRT-PCR analysis demonstrated an increased expression of *UeR40* and a decreased expression of *UeGYD* and *UeGUD* in *U. esculenta* from JB plants at 25 °C compared to 35 °C (Figure 7), together with higher growth rate of *U. esculenta* in BM added with amino acids compared to control at 25 °C, suggesting that *U. esculenta* has high amino acid requirements for cell proliferation at lower temperature.

However, genomic analysis of *U. esculenta* indicated that it lacks certain genes involved in amino acid metabolism [60]. In addition to amino acid synthesis in *Z. latifolia* infected with *U. esculenta* at cooler temperature, these results led us to the hypothesis that *U. esculenta* may be capable of altering amino acid mechanism of *Z. latifolia* and absorption of amino acids from host plant cells to provide nitrogen sources for fungal cell proliferation [7]. Interestingly, the upregulation of UeMSTRG.02678 and its target gene *UeMFS* in JB plant at 25 °C compared to 35 °C suggested that uptake of several amino acids was highly induced in *U. esculenta* under lower temperature [61]. Moreover, DE genes encoding amino acid transporters were identified in *U. esculenta* from JB plant at 25 °C compared to 35 °C (Appendix A), implying more active exchange of substances between *U. esculenta* and *Z. latifolia* during culm gall formation under lower temperature.

Previous studies have demonstrated that fungus can influence sugar metabolism and transport in plants to facilitate absorption of sugars from host plants and to promote the cell proliferation during the gall formation [38,62]. In *Z. latifoa* plants, *U. esculenta* infection enhance photosynthesis of host plants and induce the synthesis of sugars [2]. Among the *cis*-target genes of lncRNA grouped in G1 and G3 (Appendix A), as well as lncRNA uniquely expressed in JB plant at 25 °C, we identified genes involved in sugar metabolism (Appendix A), indicating that temperature variation may affect the sugar metabolism of *Z. latifolia* infected with *U. esculenta*. However, high temperature causes a significant reduction in photosynthesis in *Z. latifolia* [6], and this may limit the supply of plant sugar to fungal cells and thus repress fungal cell proliferation.

To summarize, our results indicate that high temperatures suppress the synthesis of amino acids and plant sugars in *Z. latifolia*, which further suppress the fungal cell proliferation of *U. esculenta*. Moreover, lower temperature alters expression of lncRNAs and genes in *U. esculenta*, this further regulate the synthesis of amino acids and plant sugars in *Z. latifolia*, which play important roles in providing plant cell nutrients required for cell division and growth. Alternatively, galls may serve as mobilising sinks to drive nutrients (amino acids, plant sugars), thereby improving the growth environment inside infected plant tissues allowing *U. esculenta* to absorb and utilise nutrients for efficient proliferation [7,62,63]. Plant cell division and fungal proliferation subsequently results in culm gall formation under lower temperature. Notably, it appears that some amino acids are dispensable for tumour formation [18]. However, little is known about the amino acid metabolism in fungi during the plant tumour formation, and their roles in plant-microbe-environment interactions. Therefore, further investigations are required to understand how amino acids participate in gall formation in *Z. latifolia*, as well as the substances exchange between *Z. latifolia* and *U. esculenta*. Based on our results, we created a model illustrating the potential mechanisms involved in temperature-dependent culm gall formation (Figure 8).

In conclusion, in *Z. latifolia* plants, we identified 293 DE lncRNAs between plants infected with *U. esculenta* grown at 25 °C and 35 °C, and 149 DE lncRNAs between CK plants without *U. esculenta* infection grown at 25 °C and 35 °C, while we only identified 10 DE lncRNAs in *U. esculenta* from JB plants grown at 25 °C and 35 °C. The potential *cis*-targets of these DE lncRNAs were further predicted, among which we found, in *Z. latifolia, cis*-target genes were involved in amino acid metabolism, plant defence response, and ET signalling, and in *U. esculenta, cis*-target genes were related to amino acid transport. Also, RNA-seq data exhibited altered expression of several genes related to amino acid metabolism, plant defence response, ET metabolism, and cytokinin metabolism in *Z. latifolia*; as well as those involved in amino acid metabolism, effectors, and protein synthesis in *U. esculenta* between plants infected with *U. esculenta* grown at 25 °C and 35 °C. These lncRNAs and their potential *cis*-targets, as well as DE genes, may play importmant roles in the regulatory networks of plant-microbe-environment interactions and temperature-dependent culm gall formation.

## 4. Materials and Methods

### 4.1. Plant Growth Conditions and Sampling

*Zizania latifolia* ‘Chongjiao 1’ plants were grown at the Experimental Farm of Zhejiang University, Hangzhou, China (120.2° E, 30.3° N). Genetically identical clone pairs differing in the presence (JB) or absence (CK) of *U. esculenta* in *Z. latifolia* plants were planted in greenhouse with controlled temperatures. Culm samples collected from JB and CK plants grown at 28 °C/19 °C (day/night), with an average temperature of 25 °C, were named JB25 and CK25, respectively. Meanwhile, culm samples collected from JB and CK plants grown at 38 °C/30 °C (day/night), with an average temperature of 35 °C, were named JB35 and CK35, respectively. Other growth management was carried out in accordance with that described by Yan et al. [2] with 13.5 h illumination duration. The culms were excised, frozen in liquid nitrogen, and stored at −80 °C until RNA extraction. Culm length, weight, and perimeter were measured. Culm volume was measured by the drainage method. Sampling and experiments were performed in triplicate.

### 4.2. Fungal Strains and Culture Conditions

The *U. esculenta* strains used in this study were UeMT09 and UeMT42, isolated from JB plants. Growth media were prepared as follows: BM basic medium, including 1 g of K_2_HPO_4_, 0.5 g of MgSO_4_·7H_2_O, 0.02 g of FeSO_4_·7H_2_O, 0.5 g of KCl, and 18 g of glucose, was dissolved in 1000 mL of distilled water and autoclaved for sterilisation. Amino acids (glutamic acid, glycine, and histidine) were added to BM to a final concentration of 20 mmol/L. The medium was filtered with 0.22 μm Millipore filters.

To investigate influence of amino acids on growth of *U. esculenta* under different temperatures, in vitro culture of *U. esculenta* were conducted at 25 °C and 35 °C. Fungal strains were grown in 100 mL of YEPS medium (2% sucrose, 2% tryptone, and 1% yeast extract; pH 7.3) for 24 h at 28 °C and 200 rpm. The strains were then centrifuged (10,000× *g*, 10 min, 25 °C), and 2 mL strain suspension in 50 mL of the prepared medium supplemented with different amino acids was incubated at 200 rpm and 25 °C or 35 °C. After 0, 12, 24, 36, 48, 60, and 72 h, the absorbance of the suspension solution was determined at 600 nm. BM and potato dextrose broth (PDB) medium were prepared as control and nutritious substrates, respectively.

### 4.3. RNA Extraction, Library Construction, and Sequencing

Total RNA was isolated from each culm sample using TRIzol reagent (Sangon Biotech, Shanghai, China) according to the manufacturer’s protocol. The RNA concentrations were quantified by Agilent 2100 Bioanalyzer (Agilent Technologies, Santa Clara, CA, USA). The strand specific sequencing was then performed for the identification of lncRNAs. Total RNA was treated to remove rRNA, and transcriptome libraries were constructed using TruSeq Stranded Total RNA with Ribo-Zero Plant kit (Illumina, San Diego, CA, USA) for mRNA and lncRNAs sequencing. These libraries were run on an Illumina HiSeq X ten sequencer (Illumina), which generated paired-end reads of 150 bp.

### 4.4. Transcriptome Assembly and lncRNA Identification

Raw data were processed by removing the adaptor polluted reads and low-quality reads, trimming the reads with more than 5% N bases and removing rRNA using SOAP2 (version 2.21) [64]. The clean data were mapped to the *Z. latifolia* genome (http://ibi.zju.edu.cn/ricerelativesgd/download.php) (accessed at 24 May 2017) and *U. esculenta* [60] genomes using HISAT2 (version 2.0.4) [65]. StringTie (version 1.2.1) was used for recreating transcripts [66]. We then adopted five steps to identify lncRNAs: (1) transcripts with an overage ≥3 calculated by Cufflinks were selected; (2) known protein-coding transcripts were removed; (3) transcripts with length < 200 bp were removed; (4) transcripts that did not pass the protein-coding-score test using the Coding Potential Calculator (CPC) [67] and Coding-Non-Coding Index (CNCI) [68] were eliminated; (5) transcripts were aligned in the pfam HMM database to remove protein-coding domains by PfamScan (E value < 0.001) [69]. The flow chart on transcriptome assembly and putative lncRNA identification was shown in Appendix A.

### 4.5. Differential Expression of lncRNAs and mRNAs

The collected RNA-seq data were used to determine the expression levels of both putative lncRNAs and mRNA transcripts. We analysed the expression distribution in terms of FPKM using RNA-Seq by Expectation-Maximisation (RSEM) software (version 1.2.12) (http://deweylab.biostat.wisc.edu/rsem) with default settings. We then performed differential expression analysis using DESeq2 [70]. Genes/lncRNAs with adjusted *p*-value < 0.05 and |Log_2_fold-change| > 1 were identified as differential expression.

### 4.6. Prediction and Functional Analysis of lncRNA Targets

To predict the function of lncRNAs, their *cis*-targets were predicted. We searched for coding genes 100 kb upstream and downstream of a lncRNA as the *cis*-target gene of the lncRNA. RNAplex was further used to identify the interaction between lncRNA and *cis*-target mRNA according to the minimum free energy based on the thermodynamic structure to predict the complementary base pairing [71]. Gene functions of *cis*-target mRNA were assigned according to the best match of the alignments (E-value < 10^−5^) by BLASTp searching against the NCBI nucleotide database.

### 4.7. Quantitative Reverse-Transcription PCR

To determine the relative expression levels of selected lncRNAs and their target genes, qRT-PCR was performed using specific primers (Appendix A). Total RNA was isolated from samples with the same procedures as described for RNA extraction. Using the M-MuLV First Strand cDNA Synthesis Kit (Sangon Biotech) with random primer p(dN)_6_, RNA was reverse-transcribed. The product was used as a template for qRT-PCR, which was carried out using the StepOne apparatus (Applied Biosystems, Foster City, CA, USA) with SGExcel UltraSYBR Mixture (Sangon Biotech). The reactions were as follows: amplification for 10 min at 95 °C, followed by 40 cycles of 15 s at 95 °C, 20 s at 60 °C, and 25 s at 72 °C. *UeActin2* and *ZlActin2* were used as internal control genes in these experiments. All reactions were conducted in triplicate for both technical and biological repetitions. The 2^−ΔΔCT^ method was used to calculate relative gene expression levels [72,73].

### 4.8. Bioinformatics Analysis

Because the expression pattern of ZlRPM1 in *Z. latifolia* differed from that in *A. thaliana*, bioinformatics analysis of *ZlRPM1* was further performed. Multiple alignments of protein were performed with DNAMAN 9.0. Amino acid sequences were available from GeneBank (https://www.ncbi.nlm.nih.gov/genbank/). The accession numbers used were as follows: *Oryza brachyantha* (XP_006664891.2), *Oryza sativa* (XP_015624344.1), *Sorghum bicolor* (XP_002453451.2), *Aegilops tauschii* (XP_020194017.1), *Panicum miliaceum* (RLM58947.1), *Dichanthelium oligosanthes* (OEL23414.1), *Setaria viridis* (XP_034568495.1), *Setaria italica* (XP_004979036.1), *Zea mays* (AQK58490.1), *Arabidopsis thaliana* (AAF27008.1). Phylogenetic tree analysis was performed using the MEGA6.0 program.

### 4.9. Statistical Analysis

All data obtained were analysed using SPSS software with advanced models (SPSS 20.0, SPSS Inc., IBM, Armonk, NY, USA). Significant differences between treatments were analysed using Tukey’s multiple comparison test (*p* < 0.05). All experiments were conducted with three biological replicates.

## Figures and Tables

**Figure 1 ijms-22-06020-f001:**
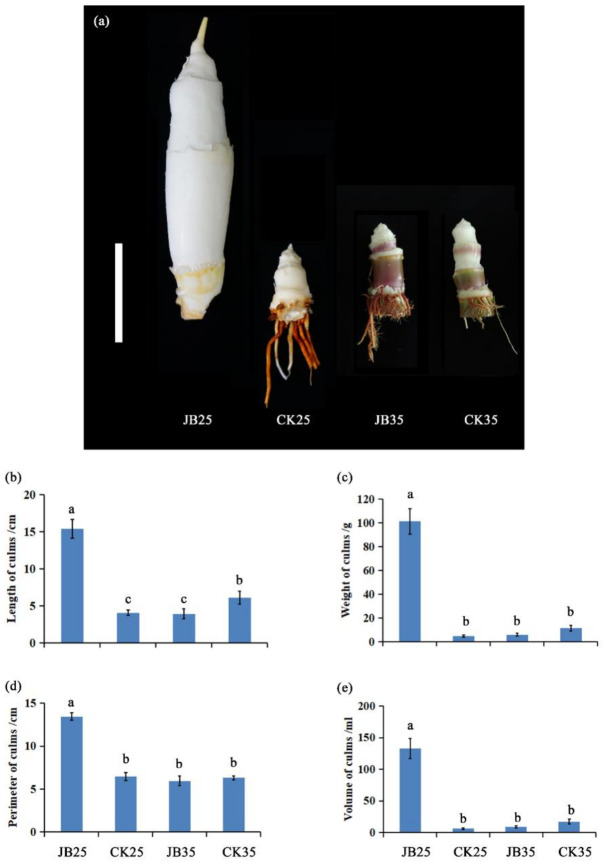
Morphology of *Zizania latifolia* culms. (**a**) Symptoms of culms; scale bar = 5 cm. (**b**) Length of culms. (**c**) Weight of culms. (**d**) Perimeter of culms. (**e**) Volume of culms. JB25 and JB35 represent culms of *Z. latifolia* plants infected with *Ustilago esculenta* grown at 25 °C and 35 °C, re-spectively. CK25 and CK35 represent culms of *Z. latifolia* plants without *U. esculenta* infection grown at 25 °C and 35 °C, respectively. Significant differences between treatments were analysed using Tukey’s multiple com-parison test (*p* < 0.05).

**Figure 2 ijms-22-06020-f002:**
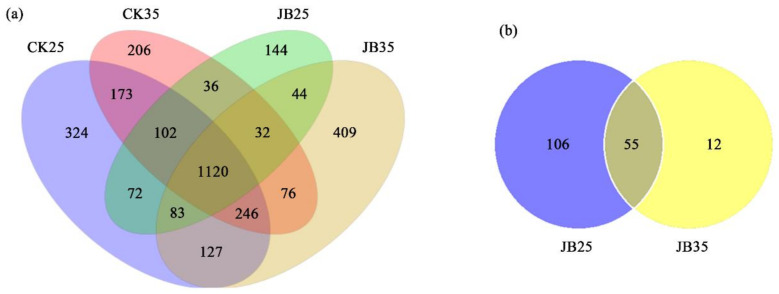
Expression of lncRNAs and mRNAs in *Zizania latifolia* and *Ustilago esculenta*. (**a**) Venn diagram showing the shared and distinct lncRNA expression in *Z. latifolia*. (**b**) Venn diagram showing the shared and distinct lncRNA expres-sion in *U. esculenta*; (**c**) Volcano map of differentially expressed lncRNAs in *Z. latifolia* between culms of *Z. latifolia* infected with *U. esculenta* grown at 25 °C (JB25) and 35 °C (JB35). (**d**) Volcano map of differentially expressed lncRNAs in *Z. latifolia* between culms of *Z. latifolia* without *U. esculenta* infection grown at 25 °C (CK25) and 35 °C (CK35). (**e**) Volcano map of differentially expressed lncRNAs in *U. esculenta* between JB25 and JB35 plants. (**f**) Venn diagram for the genes expressed in *Z. latifolia*. (**g**) Venn diagram for the genes expressed in *U. esculenta* from *Z. latifolia* plants infected with *U. esculenta*. (**h**) Volcano map of differentially expressed genes of *Z. latifolia* between JB25 and JB35 plants. (**i**) Volcano map of differentially expressed genes of *Z. latifolia* between CK25 and CK35 plants. (**j**) Volcano map of differentially expressed genes of *U. esculenta* between JB25 and JB35 plants.

**Figure 3 ijms-22-06020-f003:**
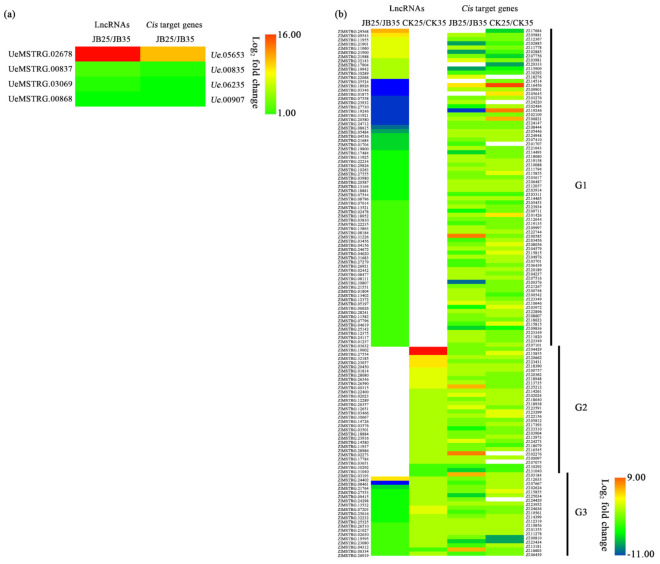
Heatmaps of differentially expressed lncRNAs and their *cis*-target genes in (**a**) *Ustilago esculenta* and (**b**) *Zizania latifolia.* JB25 and JB35 represent culms of *Z. latifolia* infected with *U. esculenta* grown at 25 °C and 35 °C, respectively. CK25 and CK35 represent culms of *Z. latifolia* without *U. esculenta* infection grown at 25 °C and 35 °C, respectively.

**Figure 4 ijms-22-06020-f004:**
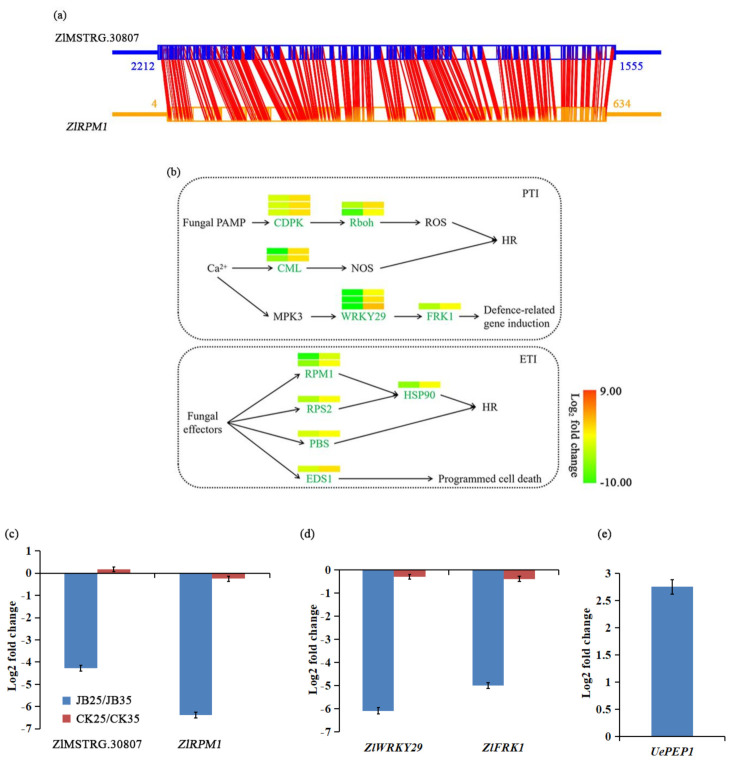
Expression analysis of lncRNA and genes related to plant defence response. (**a**) Schematic diagram showing the antisense lncRNA-mRNA interaction between lncRNA ZlMSTRG.30807 and *ZlRPM1* (disease resistance protein RPM1) gene. (**b**) Transcriptional changes of genes involved in plant defence response in *Zizania latifolia*. The relative expression levels between culms of *Z. latifolia* infected with *Ustilago esculenta* grown at 25 °C (JB25) and 35 °C (JB35) (left) and between culms of *Z. latifolia* without *U. esculenta* infection grown at 25 °C (CK25) and 35 °C (CK35) (right) are indicated using heat maps above the genes. Green indicates decreased levels, and red indicates increased levels in JB25 or CK25. The full names and the abbreviation of the transcripts are presented in Appendix A. (**c**) qRT-PCR validation for the lncRNA ZlMSTRG.30807 and its *cis*-target gene *ZlRPM1* in JB25 and JB35, and in CK25 and CK35. (**d**) qRT-PCR analysis of the pattern-triggered immunity marker genes *ZlWRKY29* (WRKY transcription factor 29) and *ZlFRK1* (leucine-rich repeat receptor-like protein kinase) in JB25 and JB35, and in CK25 and CK35. (**e**) qRT-PCR analysis of *UePEP1* (effector pep 1) gene in *U. esculenta*.

**Figure 5 ijms-22-06020-f005:**
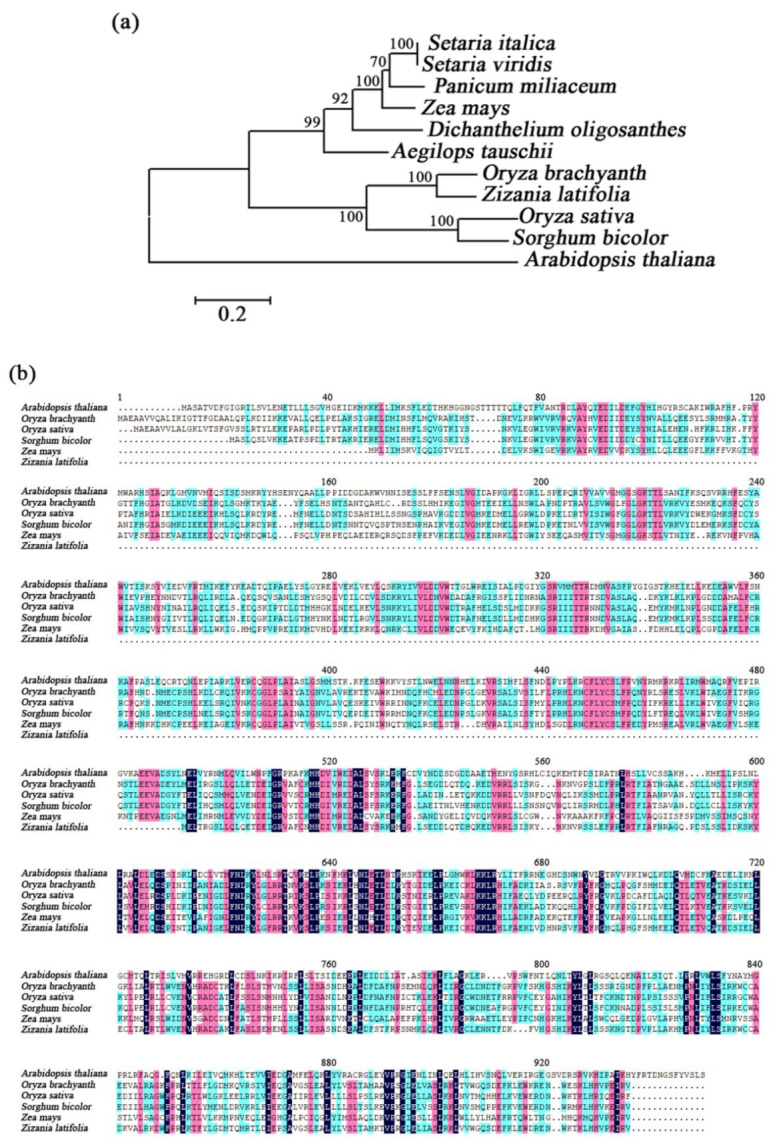
Phylogenetic analysis of *ZlRPM1*. (**a**) Phylogenetic tree analysis of *ZlRPM1*. (**b**) Amino acid sequence alignment of *ZlRPM1*.

**Figure 6 ijms-22-06020-f006:**
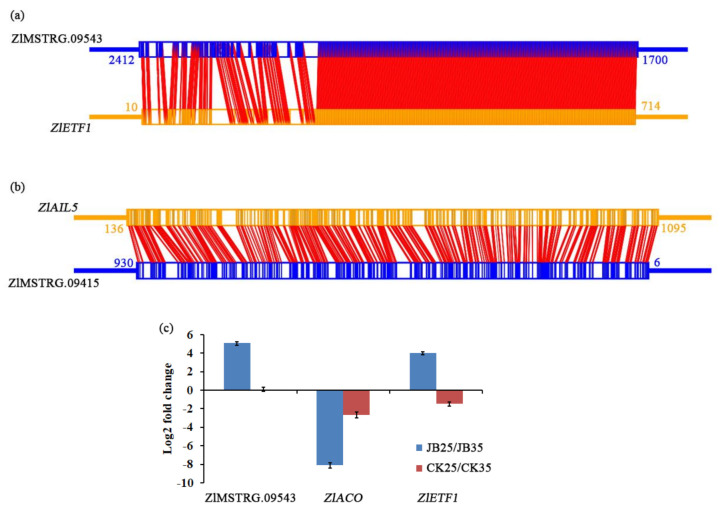
Expression analysis of lncRNA and genes related to ethylene. (**a**,**b**) Schematic diagram showing the antisense lncRNA-mRNA interaction between lncRNA ZlMSTRG.09543 and *ZlETF1* (ethylene-responsive transcription factor 1) and between lncRNA ZlMSTRG.09415 and *ZlAIL5* (AP2-like ethylene-responsive transcription factor AIL5). (**c**) qRT-PCR val-idation for the lncRNA ZlMSTRG.09543 and its *cis*-target gene *ZlETF1* and *ZlACO* (1-aminocyclopropane-1-carboxylate oxidase). JB25 and JB35 represent culms of *Zizania latifolia* infected with *Ustilago esculenta* grown at 25 °C and 35 °C, re-spectively. CK25 and CK35 represent culms of *Z. latifolia* without *U. esculenta* infection grown at 25 °C and 35 °C, respect-tively.

**Figure 7 ijms-22-06020-f007:**
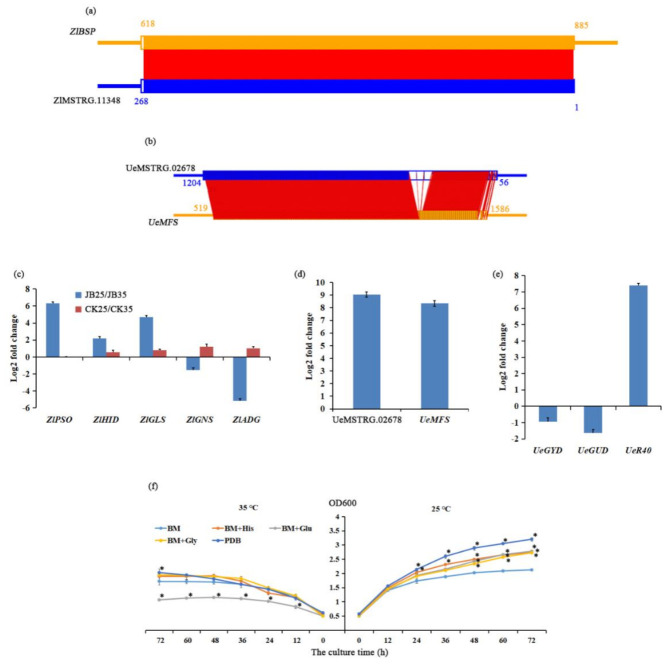
Analysis of lncRNA and genes related to amino acid metabolism. (**a**,**b**) Schematic dia-gram showing the antisense lncRNA-mRNA interaction between lncRNA ZlMSTRG.11348 and *ZlBSP* (bark storage protein) gene and between lncRNA UeMSTRG.02678 and *UeMFS* (major facil-itator superfamily transporter) gene. (**c**) qRT-PCR analysis of genes involved in amino acid metab-olism. *ZlPSO*, probable sarcosine oxidase. *ZlHID*, histidinol dehydrogenase. *ZlGLS*, glutamate synthase; *ZlGNS*, glutamine synthetase; *ZlADG*, aldehyde dehydrogenase. JB25 and JB35 represent culms of *Zizania latifolia* infected with *Ustilago esculenta* grown at 25 °C and 35 °C, respectively. CK25 and CK35 represent culms of *Z. latifolia* without *U. esculenta* infection grown at 25 °C and 35 °C, respectively. (**d**) qRT-PCR validation for the lncRNA UeMSTRG.02678 and its *cis*-target gene *UeMFS* in *U. esculenta* in JB25 and JB35 plants. (**e**) qRT-PCR analysis of genes involved in amino acid metabolism in *U. esculenta* in JB25 and JB35 plants. *UeGYD*, glycine decarboxylase; *UeGUD*, glutamate decarboxylase; *UeR40*, ribosomal 40S subunit protein. (**f**) Growth of *U. escu-lenta* under different amino acid treatments at 35 °C (left) and 25 °C (right) in vitro. * indicates sig-nificant difference between treatment and control (*p* < 0.05). BM, basic medium; His, histidine; Glu, glutamic acid; Gly, glycine; PDB, potato dextrose broth.

**Figure 8 ijms-22-06020-f008:**
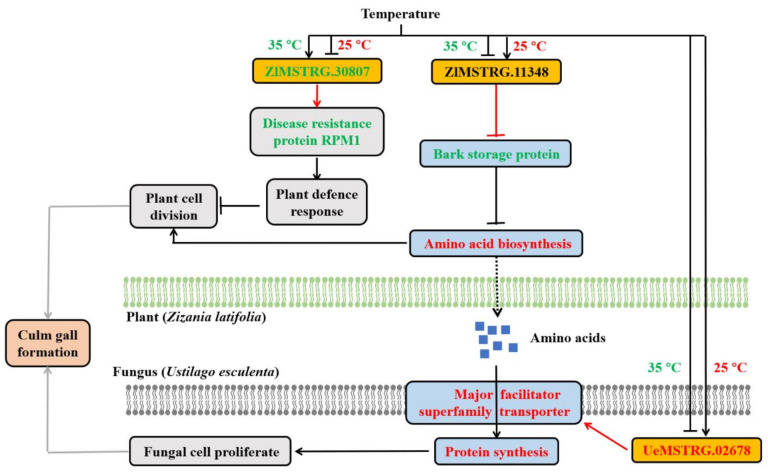
Hypothetical model of temperature-dependent culm gall formation. Under lower temperature (25 °C), the ex-pression of lncRNA ZlMSTRG.30807 is suppressed resulting in down-regulation of gene encoding disease resistance pro-tein RPM1 and inhibition of the plant defence response, which induces plant cell division in *Zizania latifolia* infected with *Ustilago esculenta*. The expression of lncRNA ZlMSTRG.11348 may be induced in *Z. latifolia* infected with *U. esculenta* and it may negatively regulate the expression of the gene encoding bark storage protein, which further induces amino acid biosynthesis under cooler temperatures. These amino acids may not only provide nutrients for plant cell division but also be obtained by *U. esculenta* through the major facilitator superfamily transporter, which is positively regulated by lncRNA UeMSTRG.02678, and further be used for protein synthesis in fungal cell proliferation. These events induce culm gall formation in *Z. latifolia* infected by *U. esculenta* under decreased temperatures. High temperature (35 °C) may inhibit the expression of lncRNA ZlMSTRG.11348 and induce the expression of lncRNA ZlMSTRG.30807 causing inhibition of amino acid metabolism and induced plant defence response in *Z. latifolia*. The expression of lncRNA UeMSTRG.02678 and its *cis*-target gene encoding major facilitator superfamily transporter is also suppressed in *U. esculenta*. These largely inhibits culm gall formation. Red and green indicate increased and decreased expression levels of genes or pathways, wherein most genes are up-regulated and down-regulated, in *Z. latifolia* infected by *U. esculenta* grown at 25 °C, respectively. Red arrows indicate the predicted regulation between lncRNAs and their *cis*-target genes.

**Table 1 ijms-22-06020-t001:** Potential *cis*-target genes of differentially expressed lncRNAs in *Ustilago esculenta* between *Zizania latifolia* plants infected with *U. esculenta* grown at 25 °C (JB25) and 35 °C (JB35) ^†^.

lncRNA ID	Log_2_FC (JB25/JB35)	*Cis*-Target Gene ID	Log_2_FC (JB25/JB35)	Annotation
UeMSTRG.02678	17.44935825	*Ue*.05653	11.40205275	Major facilitator superfamily transporter
UeMSTRG.00837	4.255446947	*Ue.*00835	2.530165481	Uncharacterised protein SPSC_00409
UeMSTRG.03069	2.581856123	*Ue*.06235	1.608946133	Hypothetical protein PSEUBRA_SCAF8g02247
UeMSTRG.00868	1.521002017	*Ue*.00907	1.211732405	Probable glutathione-dependent formaldehyde dehydrogenase

^†^ Log_2_FC, Log_2_ fold change.

**Table 2 ijms-22-06020-t002:** Differentially expressed genes involved in ethylene biosynthesis and cytokinin metabolism in *Zizania latifolia* between *Z. latifolia* infected with *Ustilago esculenta* grown at 25 °C (JB25) and 35 °C (JB35), as well as *Z. latifolia* plants without *U. esculenta* infection grown at 25 °C (CK25) and 35 °C (CK35) ^†^.

	Gene ID	Log_2_FC (JB25/JB35)	Log_2_FC (CK25/CK35)	Annotation
Ethylene biosynthesis	*Zl*.12708	NA	3.080401507	1-aminocyclopropane-1-carboxylate oxidase 1 isoform X1
	*Zl*.09313	NA	−1.126827996	1-aminocyclopropane-1-carboxylate oxidase
	*Zl*.07467	−6.974635202	−4.772315741	Putative 1-aminocyclopropane-1-carboxylate oxidase
	*Zl*.10315	−4.669026766	−1.083063024	1-aminocyclopropane-1-carboxylate synthase
	*Zl*.09320	−2.994981925	−1.715749108	1-aminocyclopropane-1-carboxylate oxidase (ACC oxidase)
	*Zl*.04853	−1.801681538	NA	S-adenosylmethionine synthetase
Cytokinin metabolism	*Zl.*15021	7.055282436	NA	Adenylate isopentenyltransferase
	*Zl.*03496	4.120206262	NA	PREDICTED: *cis*-zeatin O-glucosyltransferase 1-like
	*Zl.*19719	4.101538026	NA	Cytokinin dehydrogenase 8-like
	*Zl.*20687	2.633461018	NA	Isopentenyltransferase 8
	*Zl.*22357	1.429568818	2.365983275	Eukaryotic tRNA isopentenyltransferase
	*Zl.*09144	NA	3.54689446	Cytokinin dehydrogenase 9
	*Zl.*19009	NA	2.130198723	PREDICTED: protein G1-like4
	*Zl.*04106	NA	1.230663683	Adenylate isopentenyltransferase
	*Zl.*05044	−4.667587558	1.315840765	Putative cytokinin dehydrogenase
	*Zl.*12768	−4.113062664	1.16901907	PREDICTED: cytokinin dehydrogenase 4-like
	*Zl.*02021	−3.741466986	NA	Putative *cis*-zeatin O-glucosyltransferase
	*Zl.*14070	−1.29936594	NA	PREDICTED: adenylate isopentenyltransferase-like

^†^ Log_2_FC, Log_2_ fold change.

## Data Availability

These sequence data have been submitted to the GenBank databases under accession number SRR12834049, SRR12834047, SRR12834050, SRR12834048.

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
