# Peer review of "Role of Long Noncoding RNAs ZlMSTRG.11348 and UeMSTRG.02678 in Temperature-Dependent Culm Swelling in Zizania latifolia"

_ijms, 2021, doi:10.3390/ijms22116020_

Round 1
Reviewer 1 Report
The manuscript by Wang and Guo reports the identification of lncRNAs from Z. latifolia and U. esculenta from cum gall tissue sample. They further report the potential functions of these lncRNAs by predicting their cis-targets. However, no functional validation has been performed to support the authors claims. These days it is common to perform functional validation of lncRNAs to ascertain their functional roles rather than mere speculations. This in fact the major drawback of the study. I urge authors to perform some kind of validation experiment to supplement the conclusion of the study. If they are unable to perform such experiments, please justify it clearly.
General comments:
I suggest improving the English language usage for a better reading experience.
Introduction:
The introduction lacks logical flow and to me it is a bit all over the place. A careful reassessment is essential to smoothly introduce the topic to readers. In addition, the section describing lncRNAs should include more information to better appreciate the importance of lncRNA associated functions in plants. For instance, the authors could describe about the lncRNA classification, mechanisms of action etc. This will also address why studying lncRNAs is important.
Material and methods:
Fungal strains and treatments- It will be useful to add a phrase describing the purpose of the growth of fungal strains at different temperatures. In its present form this not clear.
Transcriptome sequencing- It is ideal to perform strand specific sequencing for the comprehensive identification of lncRNAs. Please clarify whether strand specific sequencing was performed or not? If not, what was the reason?
Transcriptome assembly and lncRNA identification- More details need to be incorporated for replicating the analysis pipeline. For instance, please add mapping parameters used. I assume that Cufflinks was used to assemble the transcripts. If this is the case, please mention this. Also, I am interested to know why a better assembler like Stringtie was not used for recreating transcripts in the study. A graphical abstract of the pipeline would be handy.
Differential expression of lncRNAs and mRNAs- Why only log2FC >1 were considered differentially expressed. What about log2FC < 1? Please clarify.
Prediction and functional analysis of lncRNA targets- Details are missing for functional analysis of lncRNA targets. Please supplement.
Quantitative reverse-transcription PCR- Please specify why random primers were used for cDNA synthesis.
Bioinformatics analysis- Please state for what purpose this was done.
Results:
Lines 105-112 Too lengthy, please shorten it, as it is confusing.
Identification and characterisation of lncRNAs in Z. latifolia and U. esculenta- Please mention the number of lncRNAs identified in this section
lncRNA and mRNA expression in Z. latifolia and U. esculenta – Line 132 “After mapping………. respectively” is not correct and misleading. You cannot identify lncRNAs by simply mapping to the genome. Please brief on the steps used for the lncRNA identification.
Lines 141-147 – It seems that this not essential to mention here, since the study is focused on identifying lncRNAs rather than mRNAs.
Functional analysis of lncRNAs – Apart from predicting the cis-targets using a window of 100Kb, whether the authors performed additional analysis to compare the expression profiles of predicted lncRNAs and their targets eg. Correlation analysis? Please clarify.
Also, please mention what functional analysis was performed to infer the putative functions of lncRNA predicted targets.
RT-qPCR analysis – How was the targets shortlisted for RT-qPCR assay?
Line 223 – I do not understand the motivation for such an analysis here. Please clarify.
Line 228 – Please avoid strong statements merely based on phylogenetic analysis. Also, I do not find any bootstrap values in the phylogram.
Discussion:
It will be more informative if the discussion section was divided into subsections. Also, please tone down affirmative statements.
Taken together, I recommend a major revision to provide authors with a chance to improve the study and manuscript.
Author Response
Point 1: The manuscript by Wang and Guo reports the identification of lncRNAs from Z. latifolia and U. esculenta from cum gall tissue sample. They further report the potential functions of these lncRNAs by predicting their cis-targets. However, no functional validation has been performed to support the authors claims. These days it is common to perform functional validation of lncRNAs to ascertain their functional roles rather than mere speculations. This in fact the major drawback of the study. I urge authors to perform some kind of validation experiment to supplement the conclusion of the study. If they are unable to perform such experiments, please justify it clearly.
Response 1: We know that it is necessary to perform the functional assay. And we tried to perform the assay, but can’t, due to no transformation system established in Z. latifolia.
General comments:
Point 2: I suggest improving the English language usage for a better reading experience.
Response 2: The manuscript is English language edited as the reviewer’s suggestion.
Introduction:
Point 3: The introduction lacks logical flow and to me it is a bit all over the place. A careful reassessment is essential to smoothly introduce the topic to readers. In addition, the section describing lncRNAs should include more information to better appreciate the importance of lncRNA associated functions in plants. For instance, the authors could describe about the lncRNA classification, mechanisms of action etc. This will also address why studying lncRNAs is important.
Response 3: We have added the lncRNA classification, mechanisms of action as the reviewer’s comments.
Material and methods:
Point 4: Fungal strains and treatments- It will be useful to add a phrase describing the purpose of the growth of fungal strains at different temperatures. In its present form this not clear.
Response 4: We have revised as the reviewer’s comments.
Point 5: Transcriptome sequencing- It is ideal to perform strand specific sequencing for the comprehensive identification of lncRNAs. Please clarify whether strand specific sequencing was performed or not? If not, what was the reason?
Response 5: Strand specific sequencing was performed. And this was added in this section.
Point 6: Transcriptome assembly and lncRNA identification- More details need to be incorporated for replicating the analysis pipeline. For instance, please add mapping parameters used. I assume that Cufflinks was used to assemble the transcripts. If this is the case, please mention this. Also, I am interested to know why a better assembler like Stringtie was not used for recreating transcripts in the study. A graphical abstract of the pipeline would be handy.
Response 6: The mapping parameters used was added in this section. Stringtie was used for recreating transcripts, which was added in this section. Cufflinks was used for further steps to identify lncRNAs as mentioned in this section (transcripts with an overage ≥ 3 calculated by Cufflinks were selected). A graphical abstract of the pipeline was added (Figure S3).
Point 7: Differential expression of lncRNAs and mRNAs- Why only log2FC >1 were considered differentially expressed. What about log2FC < 1? Please clarify.
Response 7: Actually, it is the absolute value of log2FC that used for the identification of differentially expressed lncRNA/gene. And, to our knowledge, it is very common to identify genes/lncRNAs with adjusted P-value < 0.05 and |Log2fold-change| > 1 as differentially expressed, which is widely used in many previous studies.
Point 8: Prediction and functional analysis of lncRNA targets- Details are missing for functional analysis of lncRNA targets. Please supplement.
Response 8: Details were added as the reviewer’s comments.
Point 9: Quantitative reverse-transcription PCR- Please specify why random primers were used for cDNA synthesis.
Response 9: We have performed Quantitative reverse-transcription PCR using Oligo dT Primers, but some lncRNAs could not be reverse-transcribed. Then we chose random primers and lncRNAs and mRNAs were reverse-transcribed. So, random primers were used for cDNA synthesis.
Point 10: Bioinformatics analysis- Please state for what purpose this was done.
Response 10: revised as the reviewer’s comments.
Results:
Point 11: Lines 105-112 Too lengthy, please shorten it, as it is confusing.
Response 11: revised as the reviewer’s comments.
Point 12: Identification and characterisation of lncRNAs in Z. latifolia and U. esculenta- Please mention the number of lncRNAs identified in this section
Response 12: revised as the reviewer’s comments.
Point 13: lncRNA and mRNA expression in Z. latifolia and U. esculenta – Line 132 “After mapping………. respectively” is not correct and misleading. You cannot identify lncRNAs by simply mapping to the genome. Please brief on the steps used for the lncRNA identification.
Response 13: revised as the reviewer’s comments.
Point 14: Lines 141-147 – It seems that this not essential to mention here, since the study is focused on identifying lncRNAs rather than mRNAs.
Response 14: We shorten it because the study is focused on lncRNAs and mRNAs are not that important. But we still think that it is essential to mention mRNA here, since many DEGs were mentioned in the following part (Plant defence response, Plant hormones and Amino acid metabolism).
Point 15: Functional analysis of lncRNAs – Apart from predicting the cis-targets using a window of 100Kb, whether the authors performed additional analysis to compare the expression profiles of predicted lncRNAs and their targets eg. Correlation analysis? Please clarify.
Response 15: We further performed additional analysis to identify the interaction between lncRNA and cis-target mRNA according to complemen-tary base pairing. These results were shown in Figure 4a, Figure 6a,b, Figure 7a,b.
Point 16: Also, please mention what functional analysis was performed to infer the putative functions of lncRNA predicted targets.
Response 16: we have revised as the reviewer’s comments. The details about this functional analysis were added in the section Prediction and functional analysis of lncRNA targets in Materials and Methods.
Point 17: RT-qPCR analysis – How was the targets shortlisted for RT-qPCR assay?
Response 17: For genes and lncRNAs in Z. latifolia, key genes in the pathways (such as marker genes in plant defence response) with |log2FC| > 2 were selected as candidate genes for RT-qPCR. For U. esculenta, the number of DEGs and DELs was small, and there were only four DEGs involved in amino acid metabolism and one DEG encoding effector. So, we selected two DEGs involved in glycine and glutamate metabolism as well as the DEG encoding effector pep1.
Point 18: Line 223 – I do not understand the motivation for such an analysis here. Please clarify.
Response 18: In our study, the results of qRT-PCR showed that the expression level of ZlRPM1 was lower at cool temperature (JB25) when compared to high temperature (JB35). This is different from previous study in Arabidopsis thaliana, which shows that RPM1-mediated responses are largely compromised at high temperature. We speculate that the long-term interaction with U. esculenta and long-term artificial selection may cause some mutation in ZlRPM1 accounting for the difference between Z. latifolia and A. thaliana. So, we performed the sequence alignment and phylogenetic analysis.
Point 19: Line 228 – Please avoid strong statements merely based on phylogenetic analysis. Also, I do not find any bootstrap values in the phylogram.
Response 19: This statement is based on the amino acid sequence alignment (Figure 5b), which shows that a portion of the amino acid sequence of RPM1 from Z. latifolia was lost (from 1 to 492) compared with that of RPM1 from other plant species (Arabidopsis thaliana, Oryza brachyanth, Oryza sativa, Sorghum bicolor, Zea mays). Bootstrap values were added in the phylogram (Figure 5a).
Discussion:
Point 20: It will be more informative if the discussion section was divided into subsections. Also, please tone down affirmative statements.
Response 20: We have revised as the reviewer’s comments.
Taken together, I recommend a major revision to provide authors with a chance to improve the study and manuscript.

Reviewer 2 Report
In this manuscript, the authors studied long non-coding RNA in the smut fungus Ustilago esculenta and its host Zizania latifolia. I basically like the concept of this study, because the importance of long non-coding RNA in plant-colonizing fungi is still unclear. The data presentation is a little bit boring to read, and I am feeling pity since there is no functional assay of lncRNA in fungus or plant. For example, it is unclear whether the deletion mutant of lncRNA in U. esculena can infect the host plant. Or, what happens if lncRNA of Z. latifolia is overexpressed in plant, etc...
Nevertheless, the data seems to be carefully assessed. Only a worrisome point is that I could not find source data of RNA-seq from PRJNA669195 at NCBI. Therefore, I would like the authors to comment on this point.
Author Response
Point 1: In this manuscript, the authors studied long non-coding RNA in the smut fungus Ustilago esculenta and its host Zizania latifolia. I basically like the concept of this study, because the importance of long non-coding RNA in plant-colonizing fungi is still unclear. The data presentation is a little bit boring to read, and I am feeling pity since there is no functional assay of lncRNA in fungus or plant. For example, it is unclear whether the deletion mutant of lncRNA in U. esculena can infect the host plant. Or, what happens if lncRNA of Z. latifolia is overexpressed in plant, etc...
Response 1: We know that it is necessary to perform the functional assay. And we tried to perform the assay, but can’t, due to no transformation system established in Z. latifolia.
Point 2: Nevertheless, the data seems to be carefully assessed. Only a worrisome point is that I could not find source data of RNA-seq from PRJNA669195 at NCBI. Therefore, I would like the authors to comment on this point.
Response 2: We are very sorry that there is something wrong with the accession number. At present, the source data of RNA-seq can be found at GenBank databases under accession number SRR12834049, SRR12834047, SRR12834050, SRR12834048.

Round 2
Reviewer 1 Report
The manuscript has been improved and can be accepted for publication.
Reviewer 2 Report
This is the revised manuscript submitted by Wang et al. While some part of text says like "lncRNA regulates ...", I am still feeling that it is strong statement without functional assay. Therefore, I would like them to weaken the statements.